# Analyzing the Intention of Consumer Purchasing Behaviors in Relation to Internet Memes Using VAB Model

**Hsin-Hui Lee [1], Chia-Hsing Liang [2] , Shu-Yi Liao [2] and Han-Shen Chen [1,3,*]**

[1]  Department of Health Diet and Industry Management, Chung Shan Medical University, No. 110, Sec. 1, Jianguo N. Rd., Taichung City 40201, Taiwan; f83523y@gmail.com
[2]  Department of Applied Economics, National Chung Hsing University, No. 250, Kuo Kuang Rd., Taichung 40227, Taiwan; 520gamma@gmail.com (C.-H.L.); sliao@nchu.edu.tw (S.-Y.L.)
[3]  Department of Medical Management, Chung Shan Medical University Hospital, No. 110, Sec. 1, Jianguo N. Rd., Taichung City 40201, Taiwan
\*  Correspondence: allen975@csmu.edu.tw; Tel.: +886-4-2473-0022 (ext. 12225)

**Abstract:** The proliferation of Internet has accelerated the dissemination of information, which has given birth to the term "Internet meme". Social network is one of the pivotal media in spreading an Internet meme. Marketers utilize Internet memes to carry out marketing activities to significantly improve their Internet exposure. We thus verify whether consumers generate purchase intention after being attracted to an Internet meme, as no such research prevails. We employ the value–attitude–behavior model as its theoretical core and discuss how the values formed by consumers under the impact of an Internet meme influence their purchasing behaviors through their attitudes. The participants of the study are Internet users who are habitual to checking Facebook. We adopted convenience sampling and developed 380 valid questionnaires. Structural equation modeling is applied to verify the study's hypotheses. The research results reveal that utilitarian and hedonic values influence the Purchase Intention through utilitarian and hedonic attitudes. In light of the aforementioned findings, it is suggested that marketers and relevant participants focus on the hedonic value brought by an Internet meme and design fun and witty Internet memes to attract consumers.

**Keywords:** Internet meme; VAB model; utilitarian value; hedonic value

## 1. Introduction

The proliferation of Internet has accelerated the dissemination of information, thereby creating the term "Internet meme". Shifman defined Internet meme as digital content with common features—such as online images and videos—that are created by Internet users and spread, mimicked, and modified through the Internet instantly. The term has been further used to describe objects that become viral in no time [1]. Wiggins and Bowers define Internet memes as spreadable media that have been remixed or parodied as emergent memes which are then iterated and spread online as memes [2]. Our definition of the internet meme is mostly explicated by Shifman. Nowadays, most Facebook users likely encounter a meme and/or distribute a meme daily [2]. In practice, companies use Internet memes to design advertisements because of its low costs and high dissemination rate [3]. The marketing pattern based on a meme for promoting products or services is termed as "meme marketing" [4]. Social network is one of the pivotal media that rapidly spread Internet meme, according to the statistics provided [5] by Libra and Cheetah Lab, the big data platform of Cheetah Data. Facebook (FB) ranks first among the social network applications downloaded in Taiwan, with a weekly active penetration rate of 69.77% (weekly number of active users of the app/total weekly number of active users of all Android apps in

Taiwan), weekly opening counts per person of 169.93, compared to Instagram, which ranks second, with a weekly active penetration rate of 21.03% and weekly opening counts per person of 77.03. Thus, FB dominates the social network. According to a survey conducted by the data analysis firm [6], the advertisement revenue of FB in 2016 reached USD 26.9 billion, an increase of USD 9.8 billion over 2015, making it the second largest advertisement platform worldwide. Some companies that use online groups generate enormous revenues by selling meme-related products to millions of users on the websites [7]. Many marketers have started using Internet meme as an marketing approach to attract consumers to repost and spread the word; however, lack of literature prevails on whether Internet meme marketing can effectively influence the values and attitudes of consumers and further strengthen their purchase intention, thereby leading to the motivation of this study. The study defines consumers attracted by Internet memes as those who notice Internet memes on social network and like, share, and comment on such posts, as well as purchase relevant items.

Value–attitude–behavior model (VAB) mainly discusses how value affects attitude and purchasing behavior [8]. VAB model is supported by empirical studies on a number of consuming scenarios [9] such as the choice of recreational activities [10] and shopping at malls [11]. Subjective value leads to the formation of values through consumers during the social and psychological development and affects behaviors through attitudes [12]. Babin et al. assessed and differentiated consumption value from two dimensions, namely, utilitarian and hedonic values [13]. Batra and Ahtola differentiated utilitarian and hedonic attitudes on the basis of the consumer behaviors [14]. Purchase intention refers to the possibility of consumers buying products [15]. Fishbein and Ajzen pointed out that if consumers held a positive attitude toward a product, the purchase intention comes into existence when a demand for that product prevails [16]. Mullet and Karson also believed that once consumers generated a specific attitude toward products or brands, an additional external factor would lead to their purchase intention [17]. On the basis of the aforementioned studies, it can be inferred that consumers develop a positive attitude toward products to begin with, and then comes the purchase intention.

In light of the aforementioned details, we center on the VAB model to discuss how the utilitarian and hedonics value generated by Internet memes affect the development of utilitarian and hedonic attitudes, which further affect consumers' purchase intention. The study is expected to shed light on whether Internet memes lead to actual consumption, or it is simply an eye-catching trick that yields no results.

## 2. Literature Review

### 2.1. VAB Model

In social psychology, VAB model is widely applied to the discussion and understanding of behaviors [8,18]. It interprets the impact of consumers' values on their attitudes and behaviors toward certain objects [8,19]. Tudoran et al. also stated that values indirectly affect behaviors through attitudes [20]. Jayawardhena discussed online shopping environment by engaging VAB model and revealed that a close correlation existed between the positive attitude toward online shopping and personal values such as hedonic value or self-fulfillment [21]. Thus, online shopping behaviors can be predicted using this correlation.

### 2.2. Utilitarian and Hedonic Values

Regardless of whether consumers are immersed into actual or virtual shopping experience, their interests comprise utilitarian and hedonic values [13,22–24]. Utilitarian value can be defined as an overall judgment of functional benefits and sacrifices [25,26]. In general, the utilitarian value refers to whether functions of products meet consumers' expectation [27], including economic value such as money, convenience, and saving time [28–30]. However, an increasing number of consumers pay attention to the importance of hedonic value [13,31]. Holbrook and Hirschman defined hedonic value as consumers' experience after using products [32]. The hedonic values are more personal and

subjective than utilitarian value, resulting in fun, fantasy, multisensory, and emotional aspects of shopping experience with the products [32,33]. Hedonic value focuses on the emotional or sentimental value experienced by consumers in the course of purchasing, mainly derived from fun and playfulness. In some cases, consumers do not necessarily buy something—what is more important is the joy of shopping [22].

### 2.3. Utilitarian and Hedonic Attitudes

In general, two fundamental causes prevail behind the purchasing behaviors of consumers, namely, instrumentality (utilitarian) and emotion (hedonic) [14]. Consumers who value practical interests are prone to evaluating the convenience and time-saving aspects, while consumers with hedonic motives are inclined to regard shopping behavior as a source of hedonic mentality [34]. Davis et al. suggested that consuming behaviors are driven by utilitarian and hedonic motives [35]. Consumers who focus on utilitarian attributes are inclined to convenience and saving time, while consumers who emphasize hedonic attributes regard shopping as a fun activity [34].

### 2.4. Utilitarian Value, Hedonic Value, and Attitudes

The interactivity of websites has utilitarian benefits such as saving time/energy, reducing risks, and increasing the possibility of better options [36] in addition to hedonic benefits [37]. Moreover, the interactivity of websites is believed to be able to improve consumers' attitudes toward online shops, increase viewing or re-checking of websites, and boost online purchase [38–41].

Based on these studies, we considered that the utilitarian and hedonic benefits of interactivity of websites may enhance the attitude of online purchase. Accordingly, We developed Hypotheses H1(a,b) and H2(a,b):

**Hypothesis (H1a).** *The utilitarian value of consumers who are attracted by an Internet meme has significant and positive impacts on their utilitarian attitude.*

**Hypothesis (H1b).** *The utilitarian value of consumers who are attracted by an Internet meme has significant and positive impacts on their hedonic attitude.*

**Hypothesis (H2a).** *The hedonic value of consumers who are attracted by an Internet meme has significant and positive impacts on their utilitarian attitude.*

**Hypothesis (H2b).** *The hedonic value of consumers who are attracted by an Internet meme has significant and positive impacts on their hedonic attitude.*

### 2.5. Attitude and Purchase Intention

Previous studies suggested that attitude influences behavioral intention [35,42–44]. Some studies revealed that consumers purchased products and services for utilitarian and hedonic reasons [13,14,32]. Voss, Spangenberg, and Grohmann stated that consumers' attitudes toward using and buying products encompassed utilitarian and hedonic aspects [45]. Moreover, they believed that utilitarian and hedonic attitudes comprised the process and outcome of consumption (i.e., feelings acquired from products and functions acquired from properties). Voss et al. presented that in the course of buying products, the utilitarian and hedonic attitudes of consumers exert positive impacts on behavioral intention [45]. According to the literatures, attitude can be divided into utilitarian and hedonic attitudes; moreover, attitude can have positive impacts on behavioral intention. Therefore, we developed the following hypotheses:

**Hypothesis (H3).** *The utilitarian attitude of consumers who are attracted by an Internet meme has significant and positive impacts on their purchase intention.*

**Hypothesis (H4).** *The hedonic attitude of consumers who are attracted by an Internet meme has significant and positive impacts on their purchase intention.*

## 3. Methodology

### 3.1. Research Framework

We selected the VAB as its basis. Based on the literature review, Figure 1 presents the research framework for investigating the relationships between utilitarian value, hedonic value, utilitarian attitude, hedonic attitude, and purchase intention in Taiwanese FB users. First, we investigated whether the utilitarian or hedonic value generated by consumers effects utilitarian attitude or hedonic attitude, and further investigated whether it effects the purchase intention.

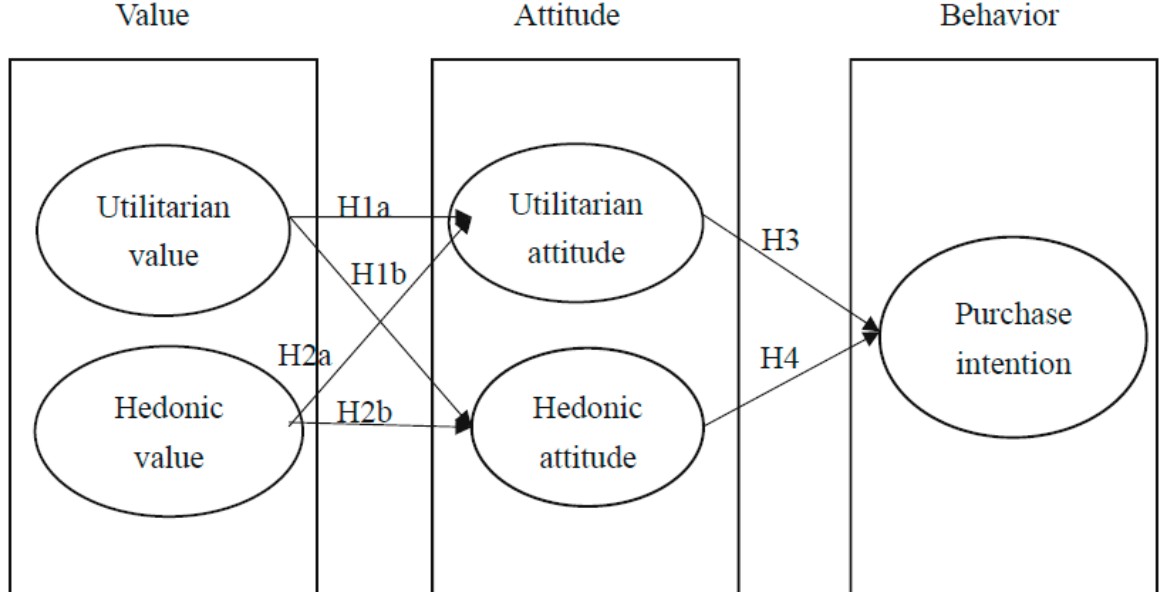

**Figure 1.** Research framework.

### 3.2. Questionnaire Design

Utilitarian value in the study is defined as the value that determines whether Internet memes create functional interests and value for consumers. The study refers to the research reports of Babin et al. and Wang et al. and designs three items for this part [13,26]. Hedonic value in this study is defined as the value that determines whether Internet memes deliver agreeable and fun experience to consumers. It refers to the research reports of Babin et al. and Wang et al. and designs five items for this part [13,26].

Utilitarian attitude in this study is defined as attitudes wherein Internet memes would lead consumers to generate functional attitude toward products. The study refers to the research reports of Voss et al. and designs three items for this part [45]. Hedonic attitude in this study is defined as the attitude wherein Internet memes would lead consumers to generate agreeable and fun attitude toward products. The study refers to the research report of Voss et al. and designs four items for this part [45]. Purchase intention of this study is defined as the free will of consumers to choose and purchase. The study refers to the research reports of Baker and Churchill, Pavlou, and Chen and Chang and designs three items for this part [46–48].

The aforementioned measuring variables have been included in the questionnaire design. Except for the demographic variables (i.e., gender, age, and education), all other variables were measured on a 5-point Likert scale consisting of strongly disagree, disagree, neither agree nor disagree, agree, and strongly agree, scoring from 1 to 5. The higher the score of an item, the more the interviewee perceives or agrees to the item.

### 3.3. Sample Size and Composition

The study chose FB users as research participants and conducted a pilot test among 50 FB users before carrying out the formal survey to ensure the questionnaire to be clear, robust, and thorough. On the basis of the results of the pilot test, a portion of the wording of the questionnaire has been amended. Regarding the formal questionnaire, the data were collected between October and December 2018 in Taiwan. Convenience sampling was adopted to conduct an online survey among FB users, although it may cause sample bias. Among the 412 responses received, 32 were deleted for excessive missing data. Thus, a total of 380 responses were used for the analysis.

We have performed sample structural analysis over 380 valid questionnaires to understand the basic information of the samples. Among the valid samples of the study, there are 218 (57%) females and 162 (43%) males; the majority (59%) of the interviewees age 21–30; and these respondents were mainly college graduates (55%).

### 3.4. Data Analysis

Data analyses conducted in this research included descriptive analyses, confirmatory factor analysis, and structural equation modeling. Descriptive statistics comprised the means and standard deviations of utilitarian value, hedonic value, utilitarian attitude, hedonic attitude, and purchase intention. Confirmatory factor analysis was performed to assess the validity of the measures using SPSS Amos version 23 (IBM, Armonk, NY, USA). The hypothesized relationships were tested using the structural equation model.

## 4. Results

### 4.1. Measurement Model: Reliability and Validity

We aim to discuss how utilitarian and hedonic values generated by Internet memes affect utilitarian and hedonic attitudes and purchase intention (Table 1). According to the result of the reliability test of each scale, the composite reliabilities of all variables are between 0.619 and 0.926; all are above 0.6 [49], which indicates good reliability of the study's questionnaire. On construct validity, for ensuring that each scale is capable of measuring the extent of the constructed theory, confirmatory factor analysis is adopted to test the suitability of fit of the measurement model. Furthermore, the average variances extracted of all variables are between 0.595 and 0.758; all are above 0.5, which indicates good convergent validity of the measurement variables in this study [49,50]. Moreover, Numally stated that if the Cronbach's $\alpha$ was larger than 0.7 [51], it suggested high reliability, while a value lower than 0.35 suggested low reliability and should be rejected. As for the part of construct validity, the factor loadings of all constructs are higher than 0.5, suggesting that the said item has a construct validity [50]. The reliabilities of the questionnaire as a whole are larger than 0.7, suggesting that the data of the questionnaires have met the criteria. Means, standard deviations, and correlations among the constructs are presented in Table 2.

**Table 1.** Results of factor loading, reliability, and validity.

| Items | Factor Loading | Cronbach's $\alpha$ | AVE | CR |
|---|---|---|---|---|
| **Utilitarian Value** | | | | |
| 1. The store that engages Internet meme on Facebook is a leading entity in its industry. | 0.603 | 0.766 | 0.595 | 0.619 |
| 2. I cannot get the information I need from the FB posts of this store (including images and videos) *. | 0.680 | | | |
| 3. I found the product I need while checking the FB posts of this store. | 0.638 | | | |
| **Hedonic Value** | | | | |
| 4. I feel happy about checking the FB posts of the store. | 0.840 | 0.829 | 0.770 | 0.837 |
| 5. I love the new FB posts of the store. | 0.883 | | | |
| 6. I just want to check the posts of the store instead of buying anything. | 0.705 | | | |
| 7. I feel very annoyed while checking *. | 0.546 | | | |
| **Utilitarian Attitude** | | | | |
| 8. Useless/Useful | 0.759 | 0.840 | 0.696 | 0.816 |
| 9. Unnecessary/Necessary | 0.744 | | | |
| 10. Impractical/Practical | 0.812 | | | |
| **Hedonic Attitude** | | | | |
| 11. Dull/Exciting | 0.862 | 0.932 | 0.758 | 0.926 |
| 12. Unhappy/Happy | 0.891 | | | |
| 13. Detest/Enjoy | 0.915 | | | |
| 14. Boring/Fun | 0.812 | | | |
| **Purchase Intention** | | | | |
| 15. I will buy products of this store. | 0.814 | 0.808 | 0.688 | 0.809 |
| 16. I might buy something from this store in the near future. | 0.804 | | | |
| 17. I will recommend products of this store to someone else. | 0.674 | | | |

Notes: * Reverse items; CR, Composite reliability; AVE, Average variance extracted.

**Table 2.** Means, standard deviations, and correlations of constructs.

| Construct | Mean | S.D. | 1 | 2 | 3 | 4 | 5 |
|---|---|---|---|---|---|---|---|
| 1. Utilitarian value | 4.27 | 0.36 | 1.00 | | | | |
| 2. Hedonic value | 4.13 | 0.62 | 0.36 ** | 1.00 | | | |
| 3. Utilitarian attitude | 3.98 | 0.57 | 0.39 ** | 0.36 ** | 1.00 | | |
| 4. Hedonic attitude | 4.31 | 0.41 | 0.37 ** | 0.28 ** | 0.37 ** | 1.00 | |
| 5. Purchase intention | 4.29 | 0.38 | 0.42 ** | 0.40 ** | 0.29 ** | 0.28 ** | 1.00 |

Note: ** $p < 0.01$.

*4.2. Structural Model and Hypothesis Testing*

AMOS version 23.0 was first used to conduct confirmatory factor analysis (CFA). Five latent constructs were contained within the measurement model (Figure 1). As shown in Table 3, the revised model exhibited an appropriate fit after CFA ($\chi^2$/df = 2.351, GFI (goodness-of-fit index) = 0.895, RMSEA (root mean square error of approximation) = 0.081, CFI (comparative fit index) = 0.928, NFI (normalized fit index) = 0.912, AGFI (adjusted goodness-of-fit index) = 0.857).

**Table 3.** Results of the fit indicators of the evaluation model.

| Fit Index | Ideal Value | Result | Conclusion |
|---|---|---|---|
| $\chi^2$/df | <3 | 2.351 | Acceptable |
| GFI | >0.9 (good fit)<br>0.8–0.89 (acceptable fit) | 0.895 | Good fit |
| AGFI | >0.9 (good fit)<br>0.8–0.89 (acceptable fit) | 0.857 | Acceptable |
| NFI | >0.9 | 0.912 | Acceptable |
| CFI | >0.9 | 0.928 | Acceptable |
| RMSEA | ≤0.05 (close fit)<br>0.05–0.08 (fair fit)<br>0.08–0.10 (mediocre fit)<br>>0.10 (poor fit) | 0.081 | Mediocre fit |

Note: GFI, goodness-of-fit index; AGFI, adjusted goodness-of-fit index; NFI, normalized fit index; CFI, comparative fit index; RMSEA, root mean square error of approximation.

We adopt structural equation modeling (SEM) to analyze the relations among the constructs of the VAB model. Figure 2 demonstrates standardized path coefficients derived from the proposed structural model: for the influence of utilitarian value on attitude, H1a ($\beta$ = 0.817, $P$ < 0.001) and H1b ($\beta$ = 0.272, $P$ < 0.001); for the influence of hedonic value on attitude, H2a ($\beta$ = 0.325, $P$ < 0.001) and H2b ($\beta$ = 0.696, $P$ < 0.001). On the basis of the aforementioned results, utilitarian and hedonic values have positive and significant impacts on attitude. Hence, H1a, H1b, H2a, and H2b are supported. Standardized regression coefficient concluded that utilitarian value has relatively greater influence on utilitarian attitude, while hedonic value has relatively greater influence on hedonic attitude.

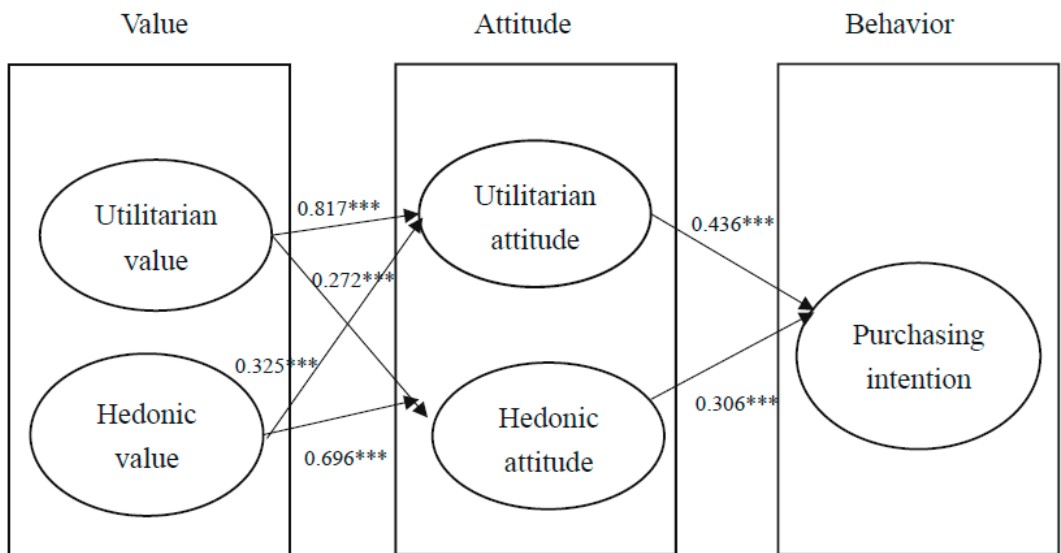

**Figure 2.** Results of the hypothesized model. *** $P$ < 0.001.

On the basis of the hypotheses of this study, utilitarian and hedonic antidotes affect purchase intention: H3 ($\beta$ = 0.436, $P$ < 0.001); H4 ($\beta$ = 0.306, $P$ < 0.001). From the above results, it is revealed that both utilitarian and hedonic attitudes have significant and positive impacts on purchase intention; therefore, H3 and H4 are supported. On the basis of the value of standardized regression coefficient, the impact of utilitarian attitude on purchase intention is slightly larger than that of hedonic attitude. A summary of the verification of the hypotheses made in this study is shown in Table 4.

<div align="center">**Table 4.** Summary of hypothesis verification.</div>

| Hypothesis | Content | Verification |
|:---:|:---:|:---:|
| H1a | The utilitarian value of consumers who are attracted by an Internet meme has significant and positive impacts on their utilitarian attitude. | Supported |
| H1b | The utilitarian value of consumers who are attracted by an Internet meme has significant and positive impacts on their hedonic attitude. | Supported |
| H2a | The hedonic value of consumers who are attracted by an Internet meme has significant and positive impacts on their utilitarian attitude. | Supported |
| H2b | The hedonic value of consumers who are attracted by an Internet meme has significant and positive impacts on their hedonic attitude. | Supported |
| H3 | The utilitarian attitude of consumers who are attracted by an Internet meme has significant and positive impacts on their Purchase Intention. | Supported |
| H4 | The hedonic attitude of consumers who are attracted by an Internet meme has significant and positive impacts on their Purchase Intention. | Supported |

## 5. Discussion

According to the aforementioned literature, utilitarian benefits of the interactivity of websites include saving time and energy, reducing risks, and possibly having better options [36]. Moreover, the interactivity of websites also brings about hedonic benefits [37]. The interactivity of websites is believed to be able to strengthen the attitude toward online store, leading consumers to browse and re-browse the website or shop online [38–41]. Thus, the study extends to the marketing approach of Internet meme adopted on social network and infers that its utilitarian and hedonic values might enhance consumers' attitude toward online stores. A portion of the validities and reliabilities of the utilitarian value are relatively low, suggesting weaker explanatory power. In light of this, the study reasons out that consumers attracted by Internet memes care less about the utilitarian value, such as saving time and reducing risks; they are more concerned about the hedonic effects of Internet memes, such as entertainment. Utilitarian and hedonic values exert significant and positive impacts on utilitarian and hedonic attitudes. Accordingly, we infer that after consumers formulate value toward Internet memes, the formulation of attitude is actually influenced. Moreover, it can be revealed that utilitarian value is more conducive to the formulation of utilitarian attitude, while hedonic value is more conducive to the formulation of hedonic attitude. Further, the possible reason for this is the identical nature of the value and attitude affecting the respective formulation.

Previous literature has revealed that the behaviors of consumers are driven by utilitarian and hedonic motives [35]. Some other scholars also have stated that when shopping, the utilitarian and hedonic attitudes of consumers have positive impacts on their behavioral intention [45]. The study extends it to the discussion of the effects of meme marketing, and the results prove that utilitarian and hedonic attitudes have positive and significant impacts on the purchase intention.

The study believes that Internet memes are indeed effective in their approach of engaging consumers, and shops mainly use witty content blended with popular current affairs to compose an Internet meme. Consumers are actually attracted to this marketing approach rather than to the products. This can also be seen in the findings of this study where the validities of utilitarian value are relatively low. In fact, it can be inferred that the utilitarian value of Internet memes is less noticed by consumers. For example, recently, Travel Frog, a phenomenal mobile game had gone viral in Japan and China, which was downloaded through App store for more than 10 million times; it had topped the free game ranking in China. The fan page of the National Palace Museum Shop made a collection of marketing graphic with frog relics featuring this trend, leading to more than 1000 reposts and an increase in the exposure of the fan page. This campaign does not aim to promote the commodities of the National Palace Museum Shop, but it is meant for branding. One can observe on the Internet that shops using Internet memes seldom focus on the value of their products; attention is paid to the

hedonic value to engage consumers, as the increase of exposure can lead more potential consumers to know about the brand.

As a result, various forms and content of memes might attract various customer groups. Hence, the study suggests that marketers should develop different forms of memes according to the relevant customer groups. For example, for products focusing on function, if a meme design is deviated to the hedonic value, it might not be able to capture the attention of the consumers who are in need of their utilitarian value. On the contrary, for the design of products or brands that are rich in hedonic value, if the campaign is limited to the introduction of their utilitarian value, it fails to attract a large number of consumers through hedonic value. The speedy permeation of Internet memes among online communities is partly due to the humorous "punchlines." The so-called punchlines can only attract consumers who understand it; otherwise, it will be ignored as useless information by consumers who do not understand them while browsing through their feeds. Therefore, it is suggested that marketers should ensure that the punchline used in their meme can be understood by the consumers they want to attract, along with guaranteeing that it is in trend. Marketers should not ignore target customers for the sake of trend, or marketing costs will be unnecessarily wasted.

Moreover, even though Internet memes do not cost a fortune, the course of designing memes can generate costs such as personnel costs. Thus, if the design is unclear, it will result in the waste of money or even in demoralizing employees because of the lackluster performance. This study can provide fact-based hints to marketers when designing Internet memes, preventing them from designing and launching new Internet memes without any considerations about the factors affecting the consumers' purchase intentions. The results suggest that consumers care less about the utilitarian value of the products or services marketed through Internet memes. Therefore, companies can pay more attention to the hedonic value of Internet memes in the process of designing and create some fun and humorous memes to attract consumers. Moreover, the memes need not introduce the products or services; however, witty Internet memes can be used to perform branding.

## 6. Conclusions and Limitations

### 6.1. Conclusions

The study's results reveal that "utilitarian value" and "hedonic value" brought by Internet memes can indeed affect the "purchase intention" of consumers through strengthening their "attitude." This conclusion supports the arguments of Voss et al. [45]. It can be concluded that after consumers are attracted to checking the posts of the store because of an Internet meme, they are even more attracted and interested in such products and end up buying something. Moreover, according to the research results, consumers who focus on the utilitarian value are more prone to increasing their purchase intention through the impact on utilitarian attitude, while consumers who focus on the hedonic value are more prone to increasing the purchase intention through hedonic attitude; this conclusion supports the argument of Kim [34]. The possible reason for this relation is that consumers who care about utilitarian value might care more about utilitarian attitude of the same nature, while consumers who care about hedonic value might care more about hedonic attitude of the same nature.

### 6.2. Limitations of the Research and Future Research

The study discusses only the utilitarian and hedonic aspects, although meme marketing is similar to viral marketing. Therefore, future research should refer to relevant influencing factors of viral marketing and further discuss what factors of Internet memes have the potential to stimulate consumers to repost, as well as how to extend the infection cycle of viral marketing, thereby allowing companies to achieve great marketing effects with low costs. Furthermore, convenience sampling adopted by this study might lead to sample bias; therefore, future research may consider referring to the E-Commerce Yearbook and adopting stratified random sampling to compare and confirm whether the findings are different as a result of any sample bias.

**Author Contributions:** Four co-authors together contributed to the completion of this article. H.-H.L. was the first author, who analyzed the data and drafted the manuscript; C.-H.L. contributed to reviewing the manuscript and revising the results and conclusion; S.-Y.L. contributed to reviewing and revising the literature, results, and conclusion; and H.-S.C. acted as corresponding author on their behalf throughout the revision and submission process.

**Funding:** This research received no external funding.

**Acknowledgments:** First, we would like to express our deepest gratitude to Li-Peng Liew, who provided the literature suggestion used in the thesis. Second, we would like to express our heartfelt thanks to all the experts who have taken the time to review this article and provide valuable comments.

**Conflicts of Interest:** The authors declare no conflict of interest.

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
