# Peer review of "Analyzing the Intention of Consumer Purchasing Behaviors in Relation to Internet Memes Using VAB Model"

_sustainability, doi:10.3390/su11205549_

Round 1
Reviewer 1 Report
The literature review / theory section of the paper gives a very general overview of some of the main concepts, but a clear theoretical grounding and building of the hypothesis based on relevant theory is missing. The current theory section includes some claims related to the definitions of intentions, attitudes and memes that this reader would have liked to read more about. Moreover, the theoretical part mentioned a review and stated claims based on the review done, but the readers need to be shown (not stated) how the review (i.e. references) show these claims to be accurate.
The description of the methods applied need to be more open, instead of stating that an appropriate process was followed; the process needs to be reported so that the readers may consider its appropriateness themselves.
Reviewer 2 Report
Basic comments:
Shifman is fine to cite for definition of memes but should also consider any of the following: Milner (2012), Wiggins & Bowers (2015) or Wiggins (2019), Knobel & Lankshear (2011) Intro can be strengthened by including Wiggins (2019) - see chapter on “commercially motivated strategic messaging” under 2.2, see literature on the so-called “experience economy” Age range of participants: 21-30 - please address their likely disposable income as a potential unknown and therefore a limitation Lines 233-234: is this not self-evident?
Reviewer 3 Report
The article is well-written. However, there are several instances of personification (writing with anthropomorphism; attributing action to objects that cannot take that action). This issue in writing makes the reading unclear.
I find it a little confusing to read a brief introduction to the use of the VAB model and a few examples of empirical studies that have used the model in the introduction. Details on the hedonic and utilitarian attitudes are also included in the Intro. The Literature Review section that follows this introduction includes a definition of the VAB model and the utilitarian and hedonic values. I suggest that these two sections, the Introduction and the Literature Review could be merged as to include: a full definition of the VAB model along with references to their previous empirical applications.
I noted that the references to previous empirical studies in the Introduction using VAB (lines. 56, 58, 59) are quite dated. I did a quick Google scholar search and I found various articles that include the VAB model. Why not update references and speak to current research on this model?
Section 3 – Includes personification almost in every sentence at the beginning of a paragraph. For instance, section 3.1 starts with a statement: “This study selected the VAB as its basis.” (Please, revise and avoid personification: the study cannot select a methodology).
I suggest that the authors provide a rationale for their choice of VAB model. The rationale can easily be completed if the authors draw upon the discussion in the Literature Review section.
Figure 1. – Research framework in line 132 is not self-explanatory. I suggest that the authors provide a brief description of the model and how they conceptualize their study.
The limitations mentioned in the end of the article require more elaboration in previous sections specially, an elaboration on convenience sample and the ways in which this sample strategy led to sample bias in the study.
Round 2
Reviewer 1 Report
In my opinion, it could be advised that the work is made focusing on the supply-side perspective in relation to Internet memes, as it is less common, and therefore it adds value to the paper.
I find the introduction particularly short… the authors should concentrate on the need of such a paper, the contribution made, objectives and outline. But the discourse meanders sometimes.
I suggest some improvements that may be made related to structure:
Section 3.1. Research Framework should be included in the end, in Section: 2.5. Attitude and Purchase Intention (that is, in Section 2. Literature review).
Author Response
請參閱附件文件。
